# Developments in Deep Learning Artificial Neural Network Techniques for Medical Image Analysis and Interpretation

**DOI:** 10.3390/diagnostics15091072

**Published:** 2025-04-23

**Authors:** Olamilekan Shobayo, Reza Saatchi

**Affiliations:** 1School of Engineering and Built Environment, Sheffield Hallam University, Pond Street, Sheffield S1 1WB, UK; r.saatchi@shu.ac.uk; 2School of Computing and Digital Technologies, Sheffield Hallam University, 151 Arundel Street, Sheffield S1 2NU, UK

**Keywords:** artificial intelligence, artificial neural networks, medical image analysis, deep learning, image classification and pattern recognition

## Abstract

Deep learning has revolutionised medical image analysis, offering the possibility of automated, efficient, and highly accurate diagnostic solutions. This article explores recent developments in deep learning techniques applied to medical imaging, including convolutional neural networks (CNNs) for classification and segmentation, recurrent neural networks (RNNs) for temporal analysis, autoencoders for feature extraction, and generative adversarial networks (GANs) for image synthesis and augmentation. Additionally, U-Net models for segmentation, vision transformers (ViTs) for global feature extraction, and hybrid models integrating multiple architectures are explored. The Preferred Reporting Items for Systematic Reviews and Meta-Analyses (PRISMA) process were used, and searches on PubMed, Google Scholar, and Scopus databases were conducted. The findings highlight key challenges such as data availability, interpretability, overfitting, and computational requirements. While deep learning has demonstrated significant potential in enhancing diagnostic accuracy across multiple medical imaging modalities—including MRI, CT, US, and X-ray—factors such as model trust, data privacy, and ethical considerations remain ongoing concerns. The study underscores the importance of integrating multimodal data, improving computational efficiency, and advancing explainability to facilitate broader clinical adoption. Future research directions emphasize optimising deep learning models for real-time applications, enhancing interpretability, and integrating deep learning with existing healthcare frameworks for improved patient outcomes.

## 1. Introduction

The use of imaging for diagnosis in healthcare is substantial, amounting to about USD 100 billion globally per year [1]. Mounting pressures on healthcare facilities and the market for imaging diagnosis have led to increasing demands for diagnostic excellence in the clinical setting due to the rising number of clinical images, greater image complexity, and faster results demanded by clinicians. As a result, the need for new technologies is centred on providing solutions that will increase the effectiveness of the clinical process, improving healthcare systems, and provide accurate diagnosis for patients while improving care quality. Therefore, there has been high demand for technologies that can aid in the automation of workflows associated with the use of medical imaging for diagnosis, leading to advances in the use of artificial intelligence (AI) methods such as deep learning to assist radiologists in analysing complex image datasets [2].

### 1.1. Deep Learning Overview

Deep learning is a subfield of machine learning that leverages artificial neural network (ANN) architecture to acquire knowledge from large datasets and perform intricate operations. One of the main advantages of deep learning techniques is their ability to mimic the information processing complexity of the human brain [2,3]. The field has been studied since the 1980s, but gained prominence in recent years. This is because of access to large datasets for model training, improved algorithm development, and increased processing power of microprocessors. The structure of an ANN is an interconnection of nodes (sometimes referred to as processing elements or neurons) that can span up to multiple layers, depending on the intricacy of the tasks and capabilities of hardware resources. Each node gathers information from the previous layer and then transmits to the subsequent layer based on the configured characteristics and set parameters. The values of the parameters used are normally initially randomised, but then iteratively updated during training based on a set learning rate [4]. A deep learning network extracts deeper and intricate information as its number of architectural layers increases, resulting in optimised performance with large datasets and training iterations. This in turn ensures precise recognition of patterns in the data [3].

### 1.2. Deep Learning in Medical Imaging, Classification, and Segmentation

Deep learning has seen extensive applications in medical image analysis. It has been used for different medical imaging modalities, including X-ray radiographs, computed tomography (CT), ultrasound (US), and magnetic resonance imaging (MRI) scans, to provide predictive diagnosis and treatment. Subtle and intricate patterns presented by these medical images are effectively identified by the adapted deep learning approach, thereby providing a means to automate the feature extraction process. Image feature extraction or selection is a process that can be performed manually by a qualified specialist, but this can be time-consuming and subjective. When deep learning models are properly trained, they have an ability to accurately identify lesion or tumours, examine membranes and tissues for differences, and undertake diverse medicine-related tasks, thereby providing accelerated diagnostic outcomes. Therefore, deep learning is emerging in the realm of medical image analysis assisting with diagnosis. A summary of deep learning applications is shown in Figure 1.

### 1.3. Challenges in Utilising Deep Learning in the Medical Field

Despite the significant advancements and capabilities of deep learning techniques in medical image analysis, there are certain limitations and challenges in their implementation and acceptance. For instance, most deep learning algorithms often lack explainability, i.e., they typically operate as black boxes [5]. In the medical field, where decision-making processes are required for diagnosis and provision of treatment, the deep learning method of convolutional neural networks (CNNs), for instance, does not provide an insight into the manner by which it came up with a decision. Other challenges that applications of deep learning techniques can face include the following

Overfitting during training. Overfitting causes poor accuracy when training deep artificial neural networks in recognising images not included during their training dataset (i.e., unseen or test images), even though the images in the training set could have been correctly recognised [6,7,8,9,10]. It has several possible causes that include insufficient training examples to facilitate adequate generalisation and excessive model parameters for the architecture. However, there are techniques that could be valuable to deal with overfitting. One technique is known as dropout [6,11,12,13,14], whereby some nodes in the architecture are temporarily left out during training. Another approach is to artificially extend the number of examples in the training dataset through a process known as data augmentation [15,16,17]. A comparison of different image data augmentation methods was reported in [18]. The technique has been applied to mammograms [19] and CT images [20]. Augmentation can be performed by manipulating the images through processes such as kernel filters, geometric transformation, random erasing, and mixing images. It can also be carried out through deep learning approaches such as adversarial training, neural style transformers, and generative adversarial networks [7]. An issue with performing data augmentation is selecting the best approach for a given set of images [21]. An exploration of the influence of different data augmentation techniques on the explainability of deep learning methods was reported in [22].Image annotation. Many deep learning algorithms are supervised, i.e., they require labelled images indicating their categories during their training phase. The labelling requires annotation of images by qualified medical practitioners. Because deep learning requires large datasets for training, this process can be time-consuming. However, there have been reports of automated and interactive image annotation methods that can assist with this operation, e.g., [23,24,25,26,27,28].Noisy images. Medical images can be noisy. Noise distorts the quality of images being used to train deep learning networks [29] and can reduce their ability to learn effectively.Interpretability. This is an area of great research interest to render decision-making by deep learning artificial neural networks more transparent, i.e., moving away from so-called black box behaviour to more interpretable decision-making. The issue of interpretability has been explored in many studies, e.g., [30,31,32,33,34,35,36]. Interpretability can be considered from multiple perspectives, e.g., user orientation for explanations provided, visualisation through graphs, charts rules etc, user comprehensibility through comprehensive reasoning, simplicity of explanations, local interpretation of a single datum and global interpretation of overall data, consistency in explanations, transparency in decision-making, and ethics and fairness in revealing bias and discrimination [31].Data sharing complexities and small datasets. Medical data gathered by a single institution may be insufficient to allow effective training of deep learning algorithms, and thus sharing across many institutions would be required. This, however, could be challenging due to regulatory, technical, and privacy concerns [37] and financial and time constraints, and limited availability of patients can limit the dataset. However, valuable techniques have been devised to address small-dataset problems in deep learning [38,39].Ethical issues. There are numerous ethical issues requiring careful consideration, e.g., patient anonymity, consent/assent, data handling and protection, data sharing, and vulnerable participants [40,41,42,43].Trust. There is an ongoing issue of relying on critical medical diagnostic results generated when the manner of their generation is not sufficiently transparent [4].Computational requirements and environmental issues. Training deep learning algorithms typically requires high computational capabilities and long durations. Many general-purpose computers do not have the means of delivering the required computational resources, and there is also the issue of the environmental aspects of using so much electricity to perform the required deep learning training [44,45].

This article explores deep learning techniques for medical image analysis, highlighting their roles in assisting diagnostic tasks. It examines CNNs for classification and segmentation, recurrent neural networks (RNNs) for sequential data, autoencoders for feature extraction, and generative adversarial networks (GANs) for data augmentation. It also discusses U-Net models for segmentation, vision transformers (ViTs) for long-range dependencies, and hybrid models that integrate multiple architectures. Different algorithms and their architectures are discussed. Associated mathematical models are presented to demonstrate the manner in which different imaging modalities operate. The study emphasises the transformative impact of deep learning in medical diagnostics and suggests future improvements in efficiency and interpretability. In the following sections, the materials and methods are explained and the results are discussed.

## 2. Materials and Methods

A systematic review was undertaken to explore the latest developments in deep learning ANNs for analysing medical images. As this field is growing, with many related articles published daily, a rapid review approach was undertaken [6]. The necessary constraints and criteria for inclusion were used when searching for literature that provided information about recent deep learning techniques used for medical image analysis. The following keywords were combined as part of the literature search: “deep learning” and “medical image analysis”. We also included imaging modalities and used “OR” and “AND” operators:

“deep learning” AND “medical image analysis”.

“deep learning” AND “medical image analysis” OR “CT”.

“deep learning” AND “medical image analysis” OR “MRI”.

“deep learning” AND “medical image analysis” OR “X-ray”.

“deep learning” AND “medical image analysis” OR “infrared thermal image”.

“deep learning” AND “medical image analysis” OR “ultrasound”.

The search was performed on different scientific databases, including Scopus, PubMed, and Google Scholar, and the inclusion and exclusion criteria are shown in Figure 2. The imaging modalities are shown in Figure 3.

We examined each article with a principal focus on the deep learning techniques deployed and the imaging modalities adopted. Articles that did not report either a deep learning algorithm or a medical imaging modality were excluded. The remaining articles were then filtered to include the deep learning techniques in whole or in part, as many works used a combination of deep learning techniques. As this is a wider review of deep learning methodologies, we have not looked critically into the clinical aspects of diseases to be diagnosed. The categories included were CNNs, RNNs, autoencoders, GANs, transfer learning (TL), ViTs, and hybrid models. We also considered studies that combined multiple deep learning algorithms, i.e., hybrid systems further refined our search to include the terms “hybrid”. For completeness, a summary of each method is included as part of the review. As CNN architectures were widely deployed, mostly with other deep learning techniques, we further narrowed the search terms to include only publications that specifically used CNNs by updating the search terms to include (“convolutional neural network” AND “medical image analysis”) OR (“CNN” AND “medical image analysis”).

Over 500 different studies were initially identified across the databases, with some duplicated across the databases, especially on PubMed and Google Scholar. After applying the exclusion criteria, this was refined to 141 articles being eventually reviewed in this study. The 141 articles were distributed over all the identified categories. We included a broad range of imaging modalities. The objective was to identify the deep learning techniques most suitable for medical imaging tasks, the processes employed to improve the monitoring and diagnosis performance of these techniques, and the optimisation performed.

## 3. Results

This section contains an overview of deep learning neural network concepts.

### 3.1. Convolutional Neural Networks

CNNs were the dominant technique for analysing medical images for diagnostic purposes [46]. It has gained immense popularity since 2012, when high-performance computing (HPC) became more accessible. This led to the ImageNet competition for different combinations of deep CNNs to achieve better diagnostic results when compared to human experts. CNNs are effective for several tasks, such as image segmentation, detection, registration, localisation, and classification [5]. They consist of numerous layers of convolutional filters with nonlinear activation functions combined with pooling layers, dropout layers, and fully connected layers. Their ability to extract complex spatial relationships and patterns in images has seen them used in various medical imaging modalities, such as magnetic resonance imaging (MRI), computed tomography (CT), X-rays, ultrasound (US), and histopathology, and more recently in infrared thermal (IRT) imaging [47,48]. Images associated with multiple conditions, e.g., bone fractures, cancer, liver diseases, pneumonia, and COVID-19, were segmented, classified, registered, and interpreted. The CNN architecture applied to the discrete Fourier-transformed infrared thermal images is shown in Figure 4. Although the study was a pilot, the model was effective in screening wrist fracture in children.

The concept of convolution is simply a mathematical operation where a filter, also known as a kernel, is applied on an input image for feature extraction. The convolution process is described mathematically next. Let the image input to the CNN be represented by a 3D matrix such that:(1)$$X\in R^{H\times W\times D}$$
where H is the height of the image, W is the width of the image, and D is the depth of the image (representing the number of channels, which is usually three for an RGB image). When applied to a set of convolutional filters for each input, the output of the convolutional filter can be represented by:(2)$${\;Z}_{i,j,k}=\sum_{h=0}^{H_{k}}\sum_{w=0}^{W_{k}}\sum_{d=0}^{D}X_{i+h,j+w,d}.K_{h,w,d,k}$$
where $Z_{i,j,k}$ is the output feature map for the filter k  at position (i,j), K is the convolutional filter (kernel), and $H_{k}\;and\;W_{k}$ are the height and width of the kernel, which are usually a sample of the size of the input. This process helps to identify local patterns such as the edges, shapes, and textures from the input image.

When the CNN algorithm is used for classification purposes, it is combined with other layers such as the activation layers, pooling layer, and the fully connected layer. The output from the convolutional filter is passed through an activation function $A_{i,j,k}$. If the rectified linear unit (ReLU) activation function is applied, the output from the convolutional layer becomes:(3)$$A_{i,j,k}=\max\;\left(0,\;Z_{i,j,k}\right)$$

This operation helps to add nonlinearity by ensuring only positive values are retained, thereby helping to learn complex patterns from the image.

The pooling layer is then applied to the output of the activation layers. The effect of the pooling operation is to reduce spatial dimensions from the convolution operation and thus help the network to capture small translations in the image. This process also helps to reduce the computational complexity of the network. Assuming the max (maximum) pooling function is applied to the activation layer with a p×p window size, the output from the pooling layer is represented by:(4)$${\;A}_{i,j,k}=\underset{h,w\in [1,p]}{\max}\;\;A_{i+h,j+w,k}$$

The fully connected layer is connected to the output of a flattened max pooling layer output. This layer takes a vector value and is a standard neural network, with each input connected to all the neurons in the next layer. The output of the fully connected (FC) layer is given by:(5)$$Z_{l}=W_{l}A_{l-1}+b_{l}$$
where $A_{l-1}$ is the input into the FC layer and $W_{l}$ and $b_{l}$ represent the weights and biases of the FC layer. The final or output layer, which is also a vector, is mostly passed through a softmax function for a classification task. This function helps to convert the output classes into probabilities with the following expression:(6)$$\hat{y_{i}}=\frac{exp\;(Z_{i})}{\sum_{j=1}^{C}exp\;(Z_{j})}$$
where $\hat{y_{i}}$ represents the probability of predicting class i, and C represents the total number of classes being differentiated. The full CNN undergoes training to correctly learn the features of the image input. This training uses a loss function, usually a cross-entropy loss, when considering a classification task. The cross-entropy loss is given by:(7)$$L=-\sum_{i=1}^{C}y_{i}\log\;\left(\hat{y_{i}}\right)$$
where $\hat{y_{i}}$ and $y_{i}$ are the predicted probabilities for class i and the target label, respectively.

The gradient of the loss function is calculated with respect to the weights through a back-propagation formula:(8)$$\frac{\partial L}{\partial W}=\frac{\partial L}{\partial Z}.\frac{\partial Z}{\partial W}$$

The weights are updated using the gradient descent, i.e.,(9)$${\;W}_{new}=W_{old}-\eta \frac{\partial L}{\partial W}$$
where η (0<η≤1) represents the learning rate controlling the convergence rate.

#### Literature Review Findings for CNNs

Most studies have used a CNN individually, though there have been studies that applied a CNN in combination with another algorithm such as UNET and GANs for medical image diagnostics. However, in this section, the focus is on a CNN application on its own (other CCN-based algorithms, such as transfer learning, are discussed in Section 3.6). A lightweight CNN algorithm was used to detect COVID-19 from chest X-ray radiograph images [49]. In that study, their proposed CNN was inspired by the ResNet model [50], in which no layers were connected in sequence, creating a skip connection whereby neurons in a particular layer can be connected to another neuron further ahead. This arrangement helped to create a lightweight model that was effective in edge detection applications. Their study was compared with other CNN models such as CVDNet and deep GRU-CNN, and showed similar performance to models with reduced computational complexity. A deeper CNN in their work was proposed that used CT scans and X-ray radiographs for the detection of several pulmonary diseases, including COVID-19 and viral pneumonia [51]. They used varied datasets consisting of different imaging modalities for their work, which provided another dimension to the efficacy of CNNs in diagnostic imaging. They proposed a 26-layer deep CNN which was inspired by a wide residual network (WRN) [52]. It provided a faster training time when considering the deep nature of the architecture. This was achievable based on the sophistication of their hardware. Their model was effective in terms of the accuracy in detecting the different pulmonary diseases when compared to traditional methods. A CNN model was used to classify ultrasonic images of fatty liver [53]. A pertinent problem in that research was the similarity in the pathological ultrasonic images used for training the CNN algorithm. This sort of challenge might pose a problem for the CNN architecture, as it may struggle to extract distinct features for different pathological images. Therefore, there will be a need for deep convolutional layers, leading to computational complexities. However, they used pixel-level feature extraction as a preprocessing step and then proposed a CNN architecture comprising two convolutional layers, a pooling layer and a fully connected layer. They also tested the proposed method with a skip connection and improved its accuracy when compared to other algorithms such as VGGNet. The authors in [54] developed a CNN model to distinguish between benign and malignant tumours using radial endobronchial ultrasound (rEBUS) images. A total of 769 images were collected from hospitals in Taiwan for model training and internal validation, with an additional 300 and 92 images from two other hospitals used for external validation. The model was trained with image augmentation techniques to improve its robustness. Internal validation showed strong performance, with an AUC of 0.88, sensitivity of 0.80, and specificity of 0.75. For external validation, the AUC was 0.76 and 0.72 at the two external sites, with varying sensitivity and specificity. After fine-tuning, performance improved, reaching AUC values of 0.78 and 0.82. The CNN also showed potential in identifying lung cancer subtypes, with moderate success for adenocarcinoma and squamous cell carcinoma, but limited accuracy for small cell lung cancer. Overall, the model proved promising for aiding rEBUS-based lung cancer diagnosis. The authors in [55] developed a deep convolutional neural network (DCNN) for the classification of ovarian cancer in the pelvic area using ultrasound images. They used data collected from ten hospitals in China between 2003 and 2019, including adult patients with adnexal (ovary-related) lesions and healthy controls. Validation of the proposed model was performed using three datasets: one internal and two external, all containing images from patients with either ovarian cancer or benign lesions. The DCNN showed high diagnostic performance, with area under the curve (AUC) scores of 0.911, 0.870, and 0.831 across the three datasets, respectively. They benchmarked their model against the diagnostic results of 35 radiologists, and the proposed DCNN model outperformed the 35 radiologists in accuracy for detecting ovarian cancer.

### 3.2. Recurrent Neural Networks

Recurrent neural networks (RNNs) are another class of neural network that have gained significant research consideration for modelling of sequential data. Their variability in length of their input and output also makes them suitable for natural language processing tasks [1]. An RNN has an internal memory unlike feed-forward networks, which helps to keep memory of the hidden states across different time stamps. In RNNs, there is a feedback loop between the outputs of the hidden layers [56]. This arrangement allows the RNNs to learn sequential patterns, making them suitable for tasks such as time-series prediction and video analysis [57]. RNN architecture normally suffers from the issue of vanishing gradients [53]. Their use transcends sequential and textual data. They can also be applied to imaging modalities, such as dynamic imaging in functional MRIs, with time-series characteristic information to monitor disease progress in a patient and to check how they respond to treatments. This functionality has been made popular by a variant of the network known as the long short-term memory network (LSTM). The LSTM works by introducing self-loops to allow for the flow of gradients for long durations [58]. The recurrent structure of the neural network is always enforced by the LSTM by introducing gating functions on the neurons in the hidden layer. The architecture of an RNN is shown in Figure 5.

The mathematical representation of an RNN is described next. Let the sequential input to the RNN at time step t  be $x_{t}$ where $x_{t}\in R^{n}$ is a vector, hence the input sequence becomes:(10)$${\;x}_{1},x_{2},\ldots ,\;x_{T}\;$$
where T represents the number of time steps in the sequence. For every time step t, a hidden state $\left(h_{t}\right)$ is maintained. Each hidden state is updated as:(11)$${\;h}_{t}=f\left(W_{hh}h_{t-1}+W_{xh}x_{t}+b_{h}\right)$$
where $W_{hh}$ represents the hidden states or recurrent weights, $W_{xh}$ is the weight of the connection between the input layer and the hidden layer, and $b_{h}$ is the bias for the hidden state. The function f represents the activation function, typically the ReLU or the hyperbolic tangent. To obtain the output of the RNN at each time step, the output from each hidden state is multiplied by its associated weights and added to a chosen bias vector. This is represented as:(12)$$y_{t}=g\left(W_{hy}h_{t}+b_{y}\right)$$
where $W_{hy}$ represents the weight matrix between the hidden state and the output and $b_{y}$ represents the output bias vector. The function g is the activation function for the output, which could be task-dependent. The softmax activation function is mostly employed for classification tasks.

#### Literature Review Findings for RNNs

RNNs are widely used for machine learning or deep learning tasks that are time-related. For image diagnostics, they can be used to monitor disease progression and patient response to treatments based on different time stamps of when the images were taken. Most studies have based their classification tasks on an RNN, typically combining it with a CNN for the feature extraction stage, as shown in Table 1.

In one study, MR images were used for classification of Alzheimer’s disease by analysing the longitudinal sequence of the MR images taken at different time steps to measure disease progression [58]. They combined a CNN for feature extraction and RNN for classification. For the RNN architecture, they cascaded three bidirectional gated recurrent units (BGRU) with the inputs from the CNN at multiple points, providing a longitudinal analysis. RNN architectures can suffer from vanishing gradients when the sequence of the images becomes too long, as they do not contain memory units to store sequences [64], hence the reason for developing variants such as LSTM and GRU. RNNs can also be used on their own without the addition of gated memory units. An RNN with slight modification was used in another study for the classification of breast cancer with microscopic histopathological imaging [58]. The RNN architecture was gain-modulated using the honey badger algorithm (HBA) by updating the weights of the RNN during training. The capacity in which RNN was used in their study was for classification of different stages of breast cancer and not disease progression, hence why gated memory approaches were not used. RNNs can also be used for image segmentation tasks, especially in cases where time-series images of a part of the body are taken, requiring sequencing of the time frame for identification of anomalies. In another study, an RNN-based LSTM model was proposed together with U-Net to label aortic MR images [60]. This sort of task uses the capacity of the RNN architecture to segment temporal images, as most annotated images normally used for classification are static. The U-Net algorithm was combined with a CNN for feature extraction, with the sequencing part achieved with the RNN algorithm. The ability of a neural network to visualise several distinct images when compared to a human makes RNNs suitable for segmenting image labels for disease identification. RNNs can also been used to denoise diagnostic images. Images used for diagnosis can be susceptible to such interference as white and salt-and-pepper noise. An LSTM model was combined with particle swarm optimisation (PSO) algorithms to optimise the batch normalisation process of the RNN training to remove noise from CT images of the lungs [59]. The technique provided improved peak signal-to-noise ratio (PSNR) when compared to traditional noise removal techniques such as filtering-based and diffusion-based techniques.

### 3.3. Autoencoders

Autoencoders are unsupervised learning models used for dimensionality reduction and feature extraction with minimal distortion when their input is compared to their output [65]. They play an important role in the deep learning paradigm for medical image analysis [66]. They can help denoise or compress medical images and are useful in anomaly detection, where unusual patterns in images indicate potential medical issues. They can also be used as a semi-supervised deep learning model to produce annotated data in situations where there is a lack of substantial amount of annotated data available for training any deep learning network for tasks such as classification or segmentation [67]. Their architecture consists of an encoder and decoder structure with a latent space to store the value of the compressed data. Both the encoder and the decoder comprise a fully connected feed-forward neural network. The encoder converts the input image into a low-dimension compressed version, which is referred to as latent space or the encoder. The latent space contains only essential features of the input from the encoder and is kept as shallow as possible in terms of the number of neurons used to retain the compressed version of the input and computational efficiency. The encoder in turn transforms the latent space to a reconstruction of the input. A loss function is generally used during training to compare the input with its reconstruction [68]. The architecture of an autoencoder with an MR image as its input is shown in Figure 6.

The mathematical representation of the dimensionality reduction function of the autoencoder’s encoder and decoder is described next. Let the input image to the encoder be represented by:(13)$$x\;\epsilon \;R^{n}$$

The encoder in turn transforms the input x into a lower-dimension latent representation:(14)$$z\;\epsilon \;R^{d}$$
where d<n.

The transformation function that produced the latent information is given as:(15)$$z=f_{enc}\left(x\right)=\sigma \left(W_{enc}x+b_{enc}\right)\;$$
where $W_{enc}\;\epsilon \;R^{d\times n}$ is the encoder weight matrix, $b_{enc}\;\epsilon \;R^{d\times n}$ represents the encoder’s bias vector, and σ represents the selected activation function of the encoder, which is commonly ReLU or a sigmoid function. The latent representation z of the input x provides compressed information by only capturing the essential features of the input image, a feature referred to as bottleneck, in which the dimension of the latent feature is smaller than the input vector. This facilitates the autoencoder to learn a better way to efficiently represent the input vectors. The decoder network tries to transform the latent space z back to the reconstructed input $\hat{x}$ to match the input vector x. The decoder function can be represented thus:(16)$$\hat{x}=f_{dec}\left(z\right)=\sigma ’(W_{dec}z+b_{dec})\;$$

The loss function is used to minimize the difference between the input x and the reconstructed input $\hat{x}$. This difference is normally quantified by binary cross-entropy error (BCE) when the dataset is binary or mean squared error (MSE) for multi-class data.

The MSE loss is computed with the following function:(17)$$L\left(x,\hat{x}\right)=\frac{1}{n}\sum_{i=1}^{n}{\left(x_{i}-{\hat{x}}_{i}\right)}^{2}$$
and the BCE loss is computed as follows:(18)$$L\left(x,\hat{x}\right)=-\sum_{i=1}^{n}[x_{i}\log\;\left({\hat{x}}_{i}\right)+\left(1-x_{i}\right)log\;(1-{\hat{x}}_{i})]\;$$

Training of the autoencoder requires finding parameters for the weights and biases of the encoder and decoder such that the reconstruction loss is minimised over the training data. Gradient-based optimisation techniques such as stochastic gradient descent (SGD) or Adam can be used for this purpose. Table 2 shows the summary of autoencoder based techniques.

#### Literature Review Findings for Autoencoders

Autoencoders are associated with both unsupervised and semi-supervised deep learning tasks. These tasks are normally preferred in the absence of the ample annotated datasets required for deep learning activities, especially in imaging analytics. Autoencoders are used to augment small annotated datasets, often before a segmentation task. For example, a method known as GenSeg was proposed that combines the generative aspects of autoencoders to generate a latent representation of tumour cells from a labelled health image and uses U-Net architecture to obtain the unique information of tumours present in the MR images. There is also other related research [69]. Noise reduction is another task that has benefited from the generative features of the autoencoder framework. The fusion of Bayes shrinkage fused wavelet transform (BSbFWT) was proposed for noise removal and an autoencoder block for generating a noiseless variant of an MR image of prostate cancer [72]. MR images can be prone to Gaussian and Rician noise, which is introduced during image capture by the MRI device and the imaging environment. The effectiveness of noise-reduced images was measured using several parameters, such as peak signal-to-noise ratio (PSNR), mean squared error (MSE), structural similarity index metric (SSIM), and mean absolute error (MAE), which outperformed numerous traditional filtering approaches presented in their work. Autoencoders can also be used in classification tasks [70,72,74,76]. Autoencoders were used to mesh a fully convoluted network for a classification task [73]. The authors proposed a convolutional mesh autoencoder (CMA) framework for the classification of syndromic craniosynostosis (SC) of three known SC variants—Muenke, Crouzon, and Apert disease—in infants and adults using 3D computed tomography (CT) images and others with very good percentages on evaluation metrics such as sensitivity, specificity, and accuracy when compared to normal subjects. These tasks are very critical, as late and inaccurate diagnosis might prove irreversible, causing permanent damage to the brain. As indicated, an autoencoder which is infused with four convolutional layers for encoding and decoding, was used for the construction of face models from the CT images. Autoencoders were also used to detect complex anomalies present in medical imaging [73,75,77]. In another study, autoencoders were applied to chest X-ray and digital pathology images [76]. Abnormalities that were barely visible, such as metastases in lymph nodes, always proved difficult to detect as they resemble normal images in pathological slides. The authors proposed a deep perpetual autoencoder that learnt the shared patterns of normal images and content similarities to abnormal ones and restored them correctly. For evaluation of their mode, they used the receiver operating characteristic (ROC), as it integrates the classification performance of the normal and the abnormal class. Their model was also evaluated on non-medical images and performed well in comparison, indicating the suitability of their autoencoder for medical imaging purposes.

### 3.4. Generative Adversarial Networks

Generative adversarial networks (GANs) are groups of deep learning artificial neural networks that can be used for generating synthetic medical images, data augmentation, and improving image resolution. They are valuable for enhancing small datasets, which are common in medical imaging, and can also be used to create better training data for improving model performance. A GAN uses an unsupervised learning algorithm and can be used for mostly semi-supervised and unsupervised learning [78,79]. A GAN consists of two main parts—the generator (G) and the discriminator (D). The generator, which comprises a multilayer perceptron (LP), learns the data distribution of the input image and produces a similar image to the input, also known as “fake data”. The job of the discriminator, which is also an MLP, is to discriminate between the generated image from the generator network and the input image. The result of this discrimination, which constitutes an error between the original input image and the generated image, is fed back to the generator input to make the generated image more realistic, i.e., closer to the original image. During training, the weights of the generator and discriminator are alternately updated, and the weight updates of the generator come from the discrimination error. Both networks are engaged in a competing optimisation process. This process continues until there is an equilibrium between the generator and the discriminator networks [1,80,81]. The architecture of a GAN is shown in Figure 7.

The generator (G) consists of an input noise vector z such that $z\sim p_{z}\left(z\right),$ where $p_{z}$ is a prior distribution that could be Gaussian. The output of the generator, which could be an image, is represented by $G\left(z\right),$ and the objective is to maximize $log\;(D\left(G\left(z\right)\right))$. The input to the discriminator (*D*) is the image data sample x and the output of the generator network. The output to the discriminator is the probability that x is real or fake. This is represented by $D\left(x\right)\epsilon \left[0,1\right],$ where 0 represents fake and 1 represents real. The goal of the discriminator network is to maximize $log(D\left(x\right)+log\left(1-D(G\left(z\right)\right)))$. The discriminator loss $L_{D}$ is given by:(19)$$L_{D}=-E_{x\sim p_{data}}\left[\log D\left(x\right)\right]-E_{x\sim p_{z}}\left[\log\left(1-\left(G\left(z\right)\right)\right)\right]$$
where $-E_{x\sim p_{data}}\left[\log D\left(x\right)\right]$ maximises the probability of correctly classifying the real data and $E_{x\sim p_{z}}[\log\left(1-D\left(G\left(z\right)\right)\right)$ minimises the probability of misclassifying the generated data. When both generator and discriminator are at a stable equilibrium, the combined minimax game function is defined as:(20)$$\underset{G}{\mathrm{Min}}\;\underset{D}{\max}\;E_{x\sim p_{data}}\left[\log D\left(x\right)\right]+E_{x\sim p_{z}}[\log\left(1-\left(G\left(z\right)\right)\right)]$$

Table 3 shows the summary of GAN-based techniques.

#### Literature Review Findings for GANs

Deep learning technique such as GANs can be used in medical imaging in two different ways. The generative network in GANs has been used for image synthesis purposes to generate synthetic datasets for tasks where there are limited annotated datasets and using the discriminating networks of GANs for anomaly detection [79]. The quality of the generated synthetic images can be measured by the use of qualitative metrics such as Fréchet inception distance (FID), which is a measure of similarities between the representations of the generated and the real input images, structural similarity index measure (SSIM), which indicates the similarities of the structures (usually image contrast and brightness), and peak signal-to-noise ratio (PSNR), which is used to analyse the sensitivity of the generated image. PSNR is the most important metric when dealing with medical images. Quantitative metrics used for generated image quality include number of parameters (NoPs) to represent the total number of trainable parameters in the GAN and floating points of operations (FLoPs), which measure the cost of computation of the network [78]. Most studies that have used GANs have used them for image synthesis, image resolution, image translation, and image denoising. For example, a lightweight GAN (LEGAN) was proposed to generate high-fidelity images from MRIs and retina fundus images [79]. That technique boasts fewer parameters used in the training process and lower FID when compared to other variants of GAN such as CGAN, DCGAN, Pix2Pix, and so on. To achieve this, they used a two-stage GAN to create a coarse-to-fine paradigm, which is necessary in generating images with high sensitivity to the fine patterns of the original image. To lower the NoPs, redundancy in the convolutional kernel was eliminated by using the principal components of a normally fully ranked convolutional kernel for feature extraction. The resolution of MR images of the brain was improved by increasing the contrast using a GAN variant known as Cycle-GAN to aid in the segmentation of the images [85]. To achieve this, they used the image-to-image translation technique to create a high tissue contrast (HTC) of the real image. The attention block of the Cycle-GAN used in that study helped in focusing on a single tissue type and increasing the contrast within the tissue. Other studies have used GANs for medical image resolution, including [85]. For denoising tasks, a GAN was used for denoising X-ray images using the CGAN variant of the GAN architecture [87]. The authors customised it to deal with spatially varying noise, which is often overlooked when dealing with medical images. To achieve this, the gradients of original images were merged with noisy images to obtain conditional information for the CGAN, thereby enhancing the contrast. The convolutional layers of the generator were used in full for better feature extraction. For improved consistency between the real and fake images, the reconstruction loss was combined with WGAN loss to create an objective loss for the network, obtaining remarkable PSNR and SSIM performance when compared to other state-of-the-art (SOTA) GAN architectures.

### 3.5. U-Net

U-Net architecture combines the best of CNN and encoder–decoder models, specifically for the purposes of segmenting medical images [91,92]. They have found application in major medical imaging modalities such as CT, X-ray, and MRI. U-Net’s ability to exploit small, annotated data samples (based on its fully connected layers) by leveraging data augmentation and improved feature extraction made it a valuable technique for medical image segmentation [92]. The U-shaped architecture with skip connections help to delineate objects in images, making it highly effective in medical image analysis, particularly in tasks like tumour detection, organ delineation, and segmentation of medical images from various modalities, and variants have been widely embraced among the many different deep learning networks [93].

U-Net architecture is mainly composed of two paths. The first path is referred to as the contracting or encoder path. It uses a downsampling module that consists of several repeating convolutional blocks for semantic and contextual feature extraction. Each convolutional block has two successive 3 × 3 convolutions, ReLU activation functions, and a pooling layer. The pooling layer serves to increase the receptive field of the convolutional network with no extra burden of computing resources that might be introduced by an additional convolutional block. The second path of U-Net is the expansive or decoder path. It is saddled with the task of upsampling spatial resolutions of the feature maps from the contracting path, usually by a factor of two. During this operation, the dimensions of the features are reduced and a pixel-wise classification/resolution score is produced. The expansive path is made up of a 2 × 2 transposed convolutional layer (reversing the operation in the contracting path), which is followed by a 3 × 3 convolutional layer and a ReLU activation function. There is also a bottleneck layer that serves as a connection between the two paths. It is also comprised of two blocks of 3 × 3 convolutional layers and a ReLU activation function. The embedded skip connections in the bottleneck copy the output of each stage of the paths, helping to learn contextual and semantic representations in the deep and shallow layers, respectively [92,94]. The U-Net architecture is shown in Figure 8.

The encoders/contracting paths shown on the left-hand side of Figure 8 comprise several layers. Each layer l in the encoder is represented by the function:(21)$$f^{l}=\sigma \left(W^{l}*f^{l-1}+b^{l}\right)$$
where $f^{l-1}$ represents input features from a previous layer, $W^{l}$, $b^{l}$ is the convolutional weights and biases, and σ represents the activation function, which is ReLU in most cases.

The output of the encoder is normally max-pooled and can be represented as:(22)$${\;f}_{pooled}^{l}=MaxPool\left(f^{l}\right)\;$$

The output is then passed through the bottleneck layer, which is another convolution function, represented by:(23)$${\;f}_{bottleneck}=\sigma \left(W_{b}*f^{L}+b_{b}\right)$$
with L representing the number of layers in the encoder/contracting path. The upsampling taking place in the decoder is performed via a skip connection at every layer on the encoder. As such, each layer in the decoder upsamples the feature map, concatenates the corresponding encoder features, and applies convolution with the given function as:(24)$$f_{upsampled}^{l}=ConvTransponse\left(f_{bottleneck}^{l}\right)\;$$(25)$${\;f}_{concat}^{l}=Concat\left(f_{upsampled}^{l},f_{encoder}^{L-1}\right)\;$$(26)$${\;f}^{l}=\sigma (W_{decoder}^{l}*f_{concat}^{l}+b_{decoder}^{l})$$

At the output layer, there is a 1 × 1 convolutional layer that uses softmax activation to map the desired output segments:(27)$$y=Softmax\left(W_{out}*f^{L_{decoder}}+b_{out}\right)$$

During training of a U-Net, the loss function used is usually the cross-entropy loss, and it is normally applied pixel-wise.

Table 4 provides a summary of works that have implemented the U-Net segmentation.

#### Literature Review Findings for U-Net

U-Net is mostly used for segmentation tasks, especially for segmentation of cancer for various imaging modalities. Over time, there have been a lot of modifications to the vanilla U-Net model as different researchers have tried to enhance different facets, ranging from the skip connections to modifying the convolutional layers with attention networks in a bid to increase segmentation quality or reduce computational burden. A multi-level feature assembly MFLA U-Net was reported that was integrated with a multi-scale information attention and pixel-vanishing attention mechanism [106]. This enhanced U-Net model was designed to boost segmentation performance. The model was tested on different medical imaging datasets with different modalities such as colonoscopy and dermoscopic images. Dice index coefficients were used as a metric to evaluate the effectiveness of the model in segmenting these images. It outperformed many state-of-the-art U-Net models on the datasets used for testing. A lightweight U-Net architecture was applied to a publicly available brain tumour dataset (BraTs) to segment brain tumours [99]. The focus of the study was mainly developing a low-resource U-Net framework that had a multimodal CNN encoder–decoder. Augmentation was excluded to reduce the computational demand on the network. The model achieved remarkable performance, with Dice coefficient values of up to 0.93 for specific segmented classes when compared to other U-Net models. An enhanced U-Net model with minimal parameters was reported [97]. The authors achieved this by developing a framework known as stack multi-connection simple reducing net, otherwise known as SRNet. This network used fewer convolution operation in the downsampling and upsampling processes, which in turn helped to reduce the total parameters of a vanilla U-Net algorithm by 20%. They also modified the original architecture by ensuring the convolutional layers were not stacked, helping to reduce information loss. Their model was also tested using the BraTs dataset. They obtained matching results with popular variants of the U-Net model, also using Dice coefficients as an evaluation tool. Others modified a vanilla U-Net for the purpose of improving the accuracy of segmentation [99]. They explored the weakness of U-Net models that only focus on contextual information and neglect other useful features of the channel. The developed HDA-ResUNet combined the best of attention mechanisms, U-Net, and dilated convolution. They evaluated their model on ISI and LiTS segmentation datasets and achieved than a conventional U-Net.

### 3.6. Transfer Learning

Deep learning techniques are known to be computationally intensive, owing to the large number of trainable parameters available in their networks. These parameters increase substantially as the network deepens. Also, the availability of large annotated medical image datasets is scarce, and this is a very important consideration in the use of deep learning for medical image analysis [107,108]. Transfer learning algorithms were developed to help solve such problems by providing a means to reparametrize an already-trained large deep learning network. These networks are CNN-based and trained on millions of images with different classes. The resulting learnt parameters are saved to be reused on other datasets. Some aspects of these networks are modified to suit new datasets. Transfer learning involves using pretrained deep learning models (e.g., ResNet, VGG, DenseNet, GogleNet, XceptionNet, AlexNet, Inception V3, and SqueezeNet) and fine-tuning them on medical images. Developing models when using transfer learning is performed in two stages: initialising the weights and fine-tuning. During initialisation of the weights, the weights of a previously trained model with a different dataset, i.e., AlexNet trained on ImageNet, as shown in Figure 9, are copied. When a new training dataset, such as medical images in this case, is used to train this model, the weights are updated. In the fine-tuning process, some of the CNN layers are frozen and thereby the weights are not being updated. Another method of fine-tuning is freezing all the CNN layers barring the classification layer that is adjusted according to the requirements of the medical image data. This technique is highly effective when the available dataset is small or specific to a particular medical condition. It reduces training time and improves diagnostic accuracy [109]. Transfer learning can be categorised into inductive, transductive, and unsupervised learning based on data labels. These can also be categorised as homogeneous and heterogeneous based on how consistent the dataset features and labels are between the source and target domain [110,111].

The concept of transfer learning can be modelled mathematically as described next. The source domain be represented as:(28) Ds=xsi,ysi for ,i=1Ns 
where $x_{s}^{\left(i\right)}\in \;X_{s\;}$ represents the input data from the source domain, $y_{s}^{\left(i\right)}\in \;Y_{s\;}$ represents the corresponding labels, and $N_{s}$ is the number of samples. The target domain is represented as:(29) DT=xTi,yTi for ,i=1NT 
where $x_{T}^{\left(i\right)}\in \;X_{T\;}$ represents the input data from the target domain and $y_{s}^{\left(i\right)}\in \;Y_{s\;}$ represents the corresponding labels. $N_{T}$ is the number of samples, with the assumption that $X_{s}\neq \;X_{T\;}$ or $Y_{s\;}\neq Y_{T\;}$. The objective of the transfer learning function is to find a model $f_{T}(x_{T};\emptyset_{T})$ that minimises the loss in the target domain. If we represent the loss function by $L_{T}$, then $L_{T}$ can be defined as:(30)$${\;L}_{T}=\frac{1}{N_{T}}\sum_{i=1}^{N_{T}}l(f_{T}(x_{T}^{\left(i\right)};\emptyset_{T}),\;y_{T}^{\left(i\right)})$$
where l is the specific loss algorithm, which can be cross-entropy or mean squared error and $\emptyset_{T}$ represents the target domain model parameters. The transfer learning process between the source and target domain takes two steps. We pretrain the model on the source domain using the objective loss function given by the equation:(31)$${\;L}_{s}=\frac{1}{N_{s}}\sum_{i=1}^{N_{s}}l(f_{s}(x_{s}^{\left(i\right)};\emptyset_{s}),\;y_{s}^{\left(i\right)})\;$$

The learned parameters $\emptyset_{s}$ from the source domain are transferred to the target domain by freezing the base layers, where $\emptyset_{s}^{base}\;is$ fixed or by fine-tuning specific layers i.e., $\emptyset_{T}^{new}$.

The target domain overall model becomes:(32)$$f_{T}\left(x_{T}^{\left(i\right)};\emptyset_{T}\right)=f_{s}^{base}\left(x_{T};\emptyset_{s}^{base}\right)+f_{T}^{new}\left(x_{T};\;\emptyset_{T}^{new}\right)$$

The learning process is then optimised by regularising the losses. Therefore, a combined loss function L is given by:(33)$$L=L_{T}+\lambda \cdot L_{regularisation}$$
where λ is the regularisation weight and $L_{regularisation}$ can be any term used for smooth transfer, e.g., $L_{2}$-norm.

#### Findings for Transfer Learning

Transfer learning (TL) techniques are mostly applied in the classification of medical images, as summarised in Table 5.

This is because the networks they are learning from have been pretrained for classification as well. Several transfer learning models, such as VGG16, DenseNet 121, and ResNet 50, were used for the binary classification tasks of X-ray radiographs and CT images of COVID-19 patients [118]. Most of the model parameters were frozen and the weights were not initialised. However, they were able to obtain very good results in terms of classification accuracy, with VGG16 performing best, with accuracy of 99%. Transfer learning models like those in [118] were compared and a custom CNN built from scratch to compare the efficacy of the TL process. The models were trained on different publicly available datasets with varying modalities, ranging from X-rays to CT scans of different diseases, including lung cancer and brain tumour. The initial convolutional layers of the TL networks used in that work was frozen and the weights of the top layers were updated. The models were trained on all the datasets, and improved accuracy was obtained with the TL models, with ResNet 50 showing the highest accuracy of 90% for the histopathological images. In another study, transfer learning techniques were trained on the ImageNet datasets [119]. The authors pretrained a novel hybrid DCNN model that combined convolutional layers with global average pooling for each layer with a skip connection in each of the convolutional layers. The 200,000 augmented unlabelled data were sourced from different repositories of biopsy breast cancer image datasets and were used to train the DCNN model. The pretrained DCNN model was then used to classify annotated skin cancer lesions as benign or malignant based on the weights from the novel pretrained model. They obtained improved accuracy when compared to other models tested on similar datasets. A transfer learning model (ResNet 18) was used for the diagnosis of different stages of Alzheimer’s disease (AD) [120]. The model was fine-tuned by unfreezing all the layers, allowing it to update all it weights based on the publicly available DICOM MRI datasets used. High accuracy was obtained for the three different classes (in the range of 99%), and their model outperformed other related works that classified AD. Qing Guan et al. [121] used an Inception-v3 TL model for the classification of papillary thyroid carcinomas (PTCs) and benign thyroid nodules using ultrasound images. A total of 2836 images from 2235 patients were used. The model was trained to crop nodule margins and make diagnostic predictions. The best performance was achieved using a 50-pixel margin and 384 × 384 image size. In the test group, Inception-v3 achieved sensitivity of 93.3% and specificity of 87.4%. Their proposed model was most accurate for nodules sized 0.5–1.0 cm.

### 3.7. Vision Transformers

Despite the significant successes recorded in enhancing the diagnostic accuracy of deep learning models such as CNN, RNN, and U-Net in the classification and segmentation of medical images, there remain some limitations. Their reliance on localised feature extraction leading to inductive bias and sequential operation makes them fall short when the medical imaging tasks requiring long-range dependency and global feature extraction [122]. Although initially designed for natural language processing tasks such as sentiment analysis [123], machine translation, and text summarisation [124], their ability to capture long range dependencies in image pixels helps to build a more robust segmentation–classification model. The vision transformer is a relatively new deep learning architecture that is increasingly being applied to medical imaging. Developed by Google in 2020, it performs segmentation or classification using the transformer architecture. The ViT creates a partition of the input images into multiple patches of 16 × 16 pixels and linearly embeds them. For the pixels to be suitable for the transformer architecture, they must be transformed into fixed-length vectors [125]. The self-attention mechanism represents the main feature of the ViT architecture, as this forms the basis of the interactions among the pixel patches. It also uses positional encoding to represent the spatial location of the image patches. Feed-forward layers placed after the self-attention layers are generally used by the model for final decisions [126]. Several large language models have been developed for medical image analysis [127]; however, we have focused on the legacy transformer model in our analysis. The original architecture of the vision transformer is shown in Figure 10.

If we represent the input to the ViT model by an image X, then X can be defined such that:(34)$$X\in \;R^{H\times W\times C}$$
where H and W represent the height and width of the image, respectively, and C is the number of channels, which is usually three for RGB images. The input image is divided into non-overlapping patches of size P such that:(35)$$X_{patches}=\left\{x_{1,}x_{2,}\ldots ,x_{N,}\right\}$$
where $N=\frac{H\times W}{P^{2}}$ is the total number of patches and each patch is flattened as:(36)$$x_{1}\in R^{P^{2}.C}$$

The patches are sent through the transformer encoder layers consisting of the multi-head attention layer, which computes the relationships between patches using the K,V,Q (keys, values, and queries) matrices, which are computed as:(37)$$\;K=Z^{\left(l-1\right)}W_{K},\;V=Z^{\left(l-1\right)}W_{V},\;Q=Z^{\left(l-1\right)}W_{Q}$$
where $Z^{\left(l-1\right)}$ is the positional patch embedding for each patch and $W_{K},\;W_{V},{and\;W}_{Q}$ are the learnable weight projection matrices, which have the same size as embedding space. This is then passed to the feed-forward neural network (FFNN) and a classification head for a classification task [128].

#### Literature Review Findings for Vision Transformers

Several studies have used ViTs and their variants, including TNT, Swin, DeiT, and PVT [129], for classification, registration, and segmentation of medical images. Just like the convolution-based transfer learning models, learnable parameters from transformer models such as those can also be used for specific DL tasks. For example, a pretrained Swin transformer was used for the classification of breast cancer using publicly available breast X-ray images [130]. The dataset was resized to fit the Swin transformer input size and augmentation was also performed to improve generalisation, achieving very high classification rates based on the selected metrics for evaluation, with accuracy of 99.9% and precision of 99.8%. The results were compared with convolution-based TL algorithms (ResNet50 and VGG16), and the Swin transformer had superior performance. MR images of ischemic strokes were classified using ViT by [131]. The vanilla ViT or ViT base used in the study also had limited fine-tuning, and hyperparameters were adjusted to fit the datasets to be classified. They also augmented and resized the MR images as preprocessing steps before deploying the ViT model. They achieved an impressive accuracy score of 97.59% when compared to the VGG16 model used in a similar study, demonstrating the superiority of transformer models for classification tasks. Transformers were used for image registration tasks [132]. In that study, the authors developed the TransMorph algorithm, which is a hybrid transformer and convolutional network. The network leveraged the encoder–decoder architecture of transformers, but instead of the attention mechanism at the decoder, this was replaced with a convolutional network. For the transformer network, they used the Swin variant due to its ability to extract feature maps at different resolutions by merging patch layers, making it suitable for the image registration task. The algorithm was tested on different image-pair datasets comprising mainly MRI and CT modalities for registration purposes. They obtained very competitive results based on Dice score evaluations when compared to both traditional and other DL methods used for similar tasks. Chuantao Wang et al. [133] proposed a ViT model with a deep neural network called ConvTrans-Net for the segmentation of lymph node tumours using ultrasound images from Kaggle. The model works by concatenating the different feature vectors of the ultrasound images that are passed on to a multilayer perceptron, and the output of the multiple attention mechanism is passed on to a feed-forward layer for segmentation purposes. Their model was evaluated using the Jaccard similarity coefficient, precision, and recall, and was benchmarked with the NCP and WS-2017 models. They obtained a Jaccard coefficient 85.21%, recall of 89.65%, and precision of 85.17 when compared to the benchmark models.

### 3.8. Hybrid Models

Most deployed deep learning methods use CNNs, as described in the preceding sections, with variants such as VGG, ResNet, LSTM, RNN, GAN, and GRU all being used for different medical image analysis. However, different authors have tried to combine the strengths of some of the models to deal with the weaknesses of the others by combining them, and this forms the basis of hybrid models [3]. Some studies have also combined different deep learning methodologies by focusing on the strengths of the methodologies. For example, a convolutional network’s strong local feature extraction has been combined with the long-range dependencies of transformers when performing medical analysis.

#### 3.8.1. Convolution-Based Hybrid Models

A hybrid model of convolution algorithms was developed comprising SegNet, MultResUNet, and the krill herd optimisation algorithm (KHO) to improve the segmentation of CT scans of liver lesions and RNA genome sequencing [134]. The SegNet framework provided the segmentation capacity of their model, utilising pixel-wise classification through the softmax layer. A CNN-based architecture was employed with MultiResUNet to handle the lesion segmentation, together with the SegNet framework. The hyperparameters of the models, when optimised through the KHO algorithm, helped to improve the segmentation process. The algorithm was tested with a publicly available LiTs dataset using evaluation metrics such as Dice coefficient, F1 score, and accuracy, and obtained better results, with F1 scores comparatively higher than the models used for comparison. A hybrid model of convolutional methods involving RestNet and U-Net (ResUNet) was developed for the segmentation of liver and tumours using CT images [135]. The model focused on improving the available models by providing improved image contrast and segmenting irregular tumour shapes and small tumours. The combination used the best of RestNet’s residual connection and U-Net’s encoder and decoder structure to enhance feature learning, segmentation precision, and efficiency. Various augmentation techniques such as rotation and reflection were also implemented to increase the variability in the dataset, and accuracy of 99.6% and a Dice coefficient of 99.2% were achieved. Yangyang Zhu et al. [136] developed a hybrid model for the classification of unexplained cervical lymphadenopathy (CLA) using ultrasound images of patients in an underdeveloped area of China. The CLA-HDM model was made up of three smaller models, each designed to handle a specific diagnosis task related to unexplained lymph node abnormalities (CLA). Model 1 checked whether the issue was benign (non-cancerous) or malignant (cancerous). Model 2 looked deeper into benign cases to decide between tuberculosis or a reactive condition. Model 3 examined malignant cases to determine if they were due to metastasis (spread from another cancer) or lymphoma. Each model had two input branches, one for greyscale ultrasound (BUS) images and another for colour Doppler (CDFI) images. The CDFI images were processed to emphasize important colour details. Then, both images were analysed by a deep learning model (ResNet-50). The efficacy of their model was evaluated using the area under the curve, which was above the 0.8 benchmark for each model. The authors in [137] developed an ensemble model of a custom-built convolutional neural network (5-CNN) and a transfer learning model using the pretrained VGG-19 architecture for the classification of thyroid disorders using ultrasound images of four categories: autoimmune, nodular, micro-nodular, and normal. The combined CNN-VGG method showed superior performance, achieving test accuracy of 97.35%, specificity of 98.43%, and sensitivity of 95.75%. It also demonstrated strong predictive capabilities, with high positive and negative predictive values, and an area under the ROC curve of 0.96.

#### 3.8.2. Convolution–Transformer-Based Hybrid Models

A hybrid model called TBConvl-NET that combines CNN, LSTM, and ViT for the segmentation of several diseases was developed using publicly available datasets of different modalities such as ultrasound and MRI [138]. The hybrid model targeted some well-known challenges in segmenting medical images, such as scale, texture, and shape of pathology. Due to the high computational resources required, they used a depth-wise separable convolution, thereby reducing computational overhead. Swin transformer blocks were used in the skip connections to help deal with the varying scales of the data and help preserve semantic information. They used the Dice index, accuracy, and Jaccard index to evaluate their model, compared the developed model with other hybrid segmentation models, and obtained improved results. A hybrid classification model that combines the transformer and the convolution model to improve the classification of skin lesion was developed using publicly available datasets [139]. It used the Swin-Unet architecture to perform image segmentation, leveraging the self-attention of the Swin transformer and robust hierarchical analysis of the UNet. This was combined with the Xception and ResNet 18 models for feature extraction to further improve on the image analysis. For hyperparameter tuning, a hybrid salp swam algorithm (HSSA) was used to obtain the optimal parameters, hence avoiding local minima during training. A gated recurrent unit (GRU) network was used for the eventual classification. Accuracy of 94.51% and 95.38% was achieved on the two datasets used in the study. The model performed better when compared to TL models like AlexNet and ResNet18. Laifa Yan et al. [140], proposed a framework focusing on the three-vessel view (3VV) that identifies three major heart vessels—the pulmonary artery, aorta, and superior vena cava—using ultrasound images. In the first step, a YOLOv5 (based on CNN architecture) model was used to detect these vessels and earmark the region of interest. In the second step, they used a modified DeepLabv3 model with a new attentional multi-scale feature fusion (AMFF) module to perform segmentation. Using a dataset of 511 images, the model achieved high accuracy, with Dice scores of over 85% for the pulmonary artery and aorta.

## 4. Discussion

In this article, we have provided a broad view of various deep learning methodologies used for medical imaging analysis. These included convolutional neural networks (CNNs), recurrent neural networks (RNNs), autoencoders, generative adversarial networks (GANs), U-Net architectures, vision transformers, and hybrid models. For each of these methodologies, some of the key findings were presented.

CNNs are extensively used for medical imaging tasks like disease detection, classification, and segmentation, and the most important contribution of the algorithms is the unrivalled feature extraction through convolutional layers, making them effective for tasks requiring spatial hierarchy. CNN models can be lightweight in terms of computing resources, as inspired by ResNet, while also balancing model performance. Deeper CNN architecture can be computationally expensive, but can be very useful in multi-disease detection, e.g., in the classification and detection of COVID-19 and viral pneumonia, achieving high accuracy due to advanced hardware. It can also be effective when the task includes pixel-level feature extraction. Examples can be seen in its use in the classification of ultrasound image classification of fatty liver. RNNs, particularly their variants like long short-term memory (LSTM) and gated recurrent units (GRUs), are suitable for sequential data tasks. Their ability to retain temporal dependencies is utilised in dynamic imaging and disease progression monitoring. RNNs, which usually have convolutional layers, are effective for classification tasks for different imaging modalities, with MRI dominating. They have been applied in the detection of several diseases, such as Alzheimer’s and breast cancer. LSTM variants of the RNN algorithm handle tasks like segmentation of temporal MR images and noise removal in diagnostic images. Some modifications have also been applied to RNN algorithm training, such as using the honey badger algorithm (HBA) for optimisation of the model training parameters.

Autoencoders are unsupervised models used for dimensionality reduction, anomaly detection, and data augmentation. Their encoder–decoder structure helps compress and reconstruct medical images with minimal distortion. Autoencoders are employed for image analysis tasks like segmentation and denoising and have been used with several imaging modalities for the analysis of different diseases. Enhancing autoencoders by fusion with wavelet transform can also help improve noise removal from medical images. They can also be combined with convolutional networks to improve disease classification. GANs are used for generating synthetic medical images, data augmentation, and improving image resolution or quality. The main components of a GAN comprise a generator for creating synthetic images and a discriminator for distinguishing between real and generated images. GANs can be very computationally intensive due to the dual deep convolutional networks. Research in this area is focused on reducing models’ parameters and maintaining performance. Different variants of GANs have employed different image analysis tasks, with each variant tuned to fit the specific tasks.

U-Nets are specialised for medical image segmentation and have been widely adopted for tasks like tumour detection and organ delineation. The architecture combines encoder–decoder pathways with skip connections for improved feature representation. Different variants of the U-Net architecture also exist to enhance the vanilla model, making them suitable for specific imaging modalities while performing the main task of segmentation. Also, enhancements have been created to reduce model operating costs while retaining the segmentation process. More recent studies have been adding transformers to U-Nets to further enhance segmentation accuracy. Transfer learning leverages pretrained models (e.g., ResNet, VGG, DenseNet) on large datasets and fine-tunes them for medical imaging tasks. They are mostly used for classification tasks. Models like VGG16 and ResNet50 perform well for COVID-19 classification, achieving very high accuracy on different imaging modalities. Some transfer learning models are quite deep, with millions of parameters, making them very computationally intensive. Custom hybrid models pretrained on unlabelled medical datasets have also been used to improve classification accuracy for skin lesions and breast cancer images, mimicking the transfer learning ideology. They provide an option in terms of computing resources and domain-specific training and learning. ViTs are a relatively newer deep learning architecture, adapted for medical imaging tasks. They require global feature extraction and long-range dependencies. They divide images into patches and process them using self-attention mechanisms. ViT models can also be pretrained like most CNN models and learning transferred for newer task. Several variants have also been developed depending on the task. With larger training datasets, ViTs have been shown to outperform CNN models when used for tasks with fewer annotated data.

Hybrid models combine the strengths of multiple architectures (e.g., CNNs with RNNs or CNNs with transformers) to improve medical image analysis. CNN-based hybrids like ResUNet combine ResNet’s feature extraction with U-Net’s segmentation capabilities. They are used widely for cancer related image segmentation. Convo-transformer hybrids like TBConvl-NET integrate CNN, LSTM, and ViT for improved segmentation and classification of diseases. Hybrid models can also be optimised. Algorithms such as the hybrid salp swarm algorithm (HSSA) could be used to obtain optimal parameters, thereby increasing model performance.

## 5. Conclusions

This review explored the transformative impact of deep learning algorithms on medical image analysis. Convolutional neural networks (CNNs) are widely used for feature extraction and disease classification across medical imaging modalities, achieving high accuracy with optimised computational costs. Recurrent neural networks (RNNs), including LSTM and GRU, enhance temporal analysis for disease progression monitoring. Autoencoders and generative adversarial networks (GANs) assist in data augmentation, denoising, and synthetic image generation. U-Net architectures improve segmentation for tumour detection and organ delineation. Vision transformers (ViTs) leverage attention mechanisms for superior classification and registration. Hybrid models combining CNNs, transformers, and optimisation techniques enhance performance, while transfer learning mitigates data scarcity, ensuring robust results across imaging applications. Together, these advancements underscore the versatility and efficiency of deep learning in medical diagnostics, paving the way for improved clinical outcomes and personalised healthcare solutions. Future advancements may focus on computational efficiency, integrating multimodal data, and enhancing interpretability for clinical adoption.

## Figures and Tables

**Figure 1 diagnostics-15-01072-f001:**
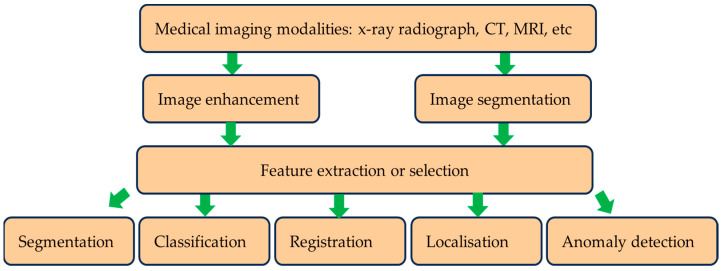
A summary of deep learning usage with medical images.

**Figure 2 diagnostics-15-01072-f002:**
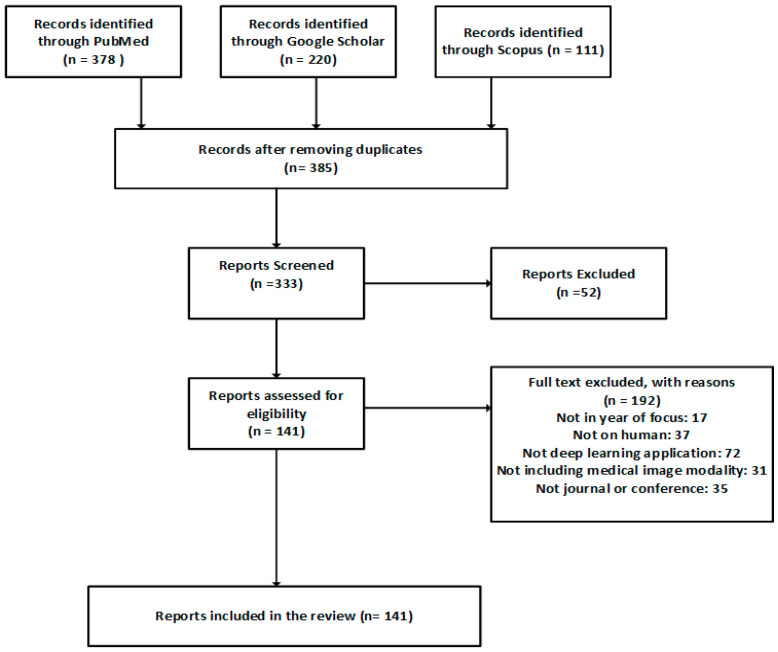
Inclusion and exclusion criteria of the systematic literature review using the PRISMA framework.

**Figure 3 diagnostics-15-01072-f003:**
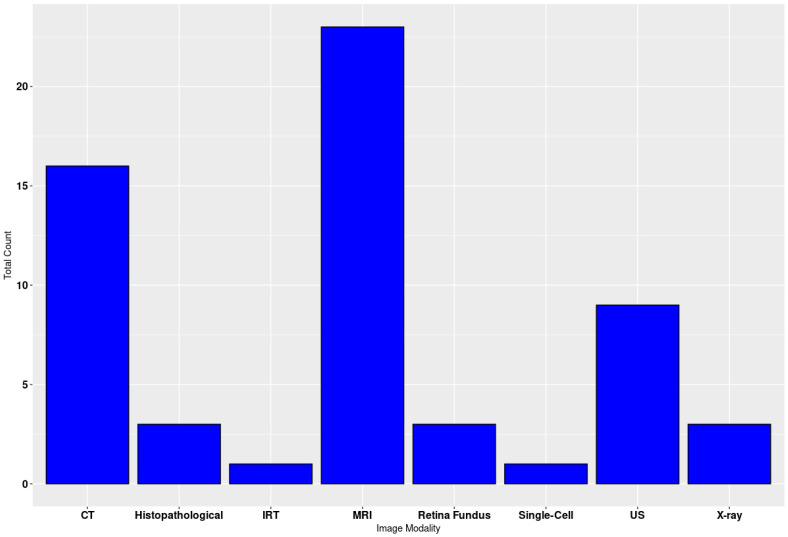
Count of imaging modalities on a section of articles included in the review.

**Figure 4 diagnostics-15-01072-f004:**
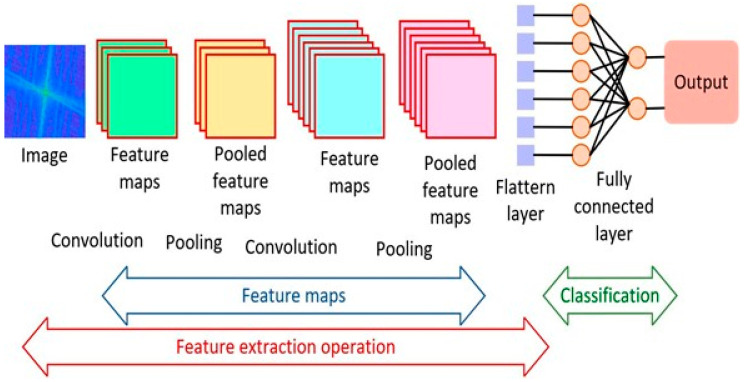
An example of a CNN used for the analysis of infrared thermal images [48].

**Figure 5 diagnostics-15-01072-f005:**
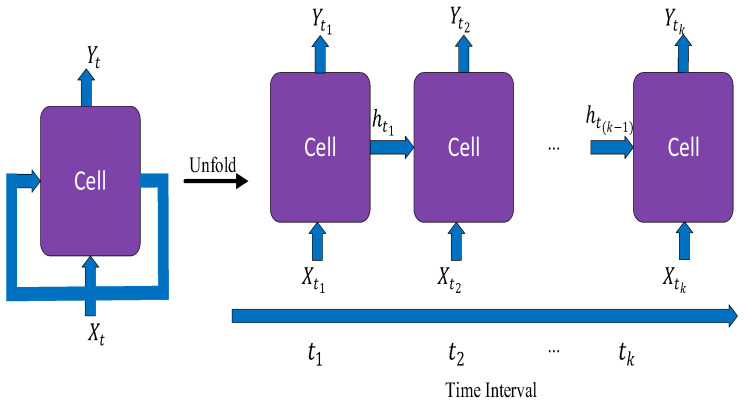
Structure of an RNN [53].

**Figure 6 diagnostics-15-01072-f006:**
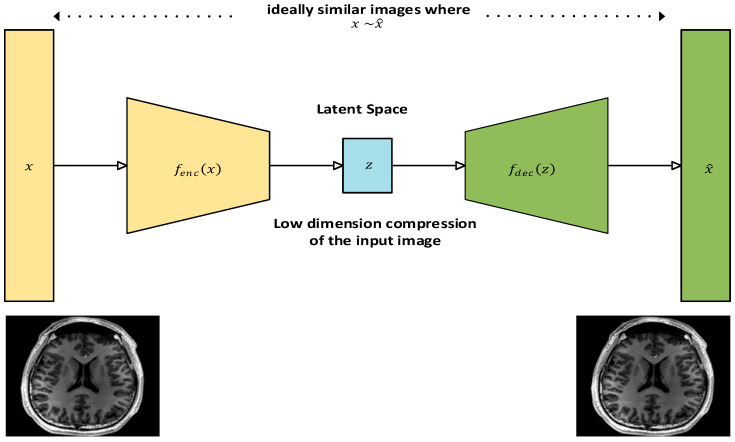
Autoencoder architecture reproducing an MR image.

**Figure 7 diagnostics-15-01072-f007:**
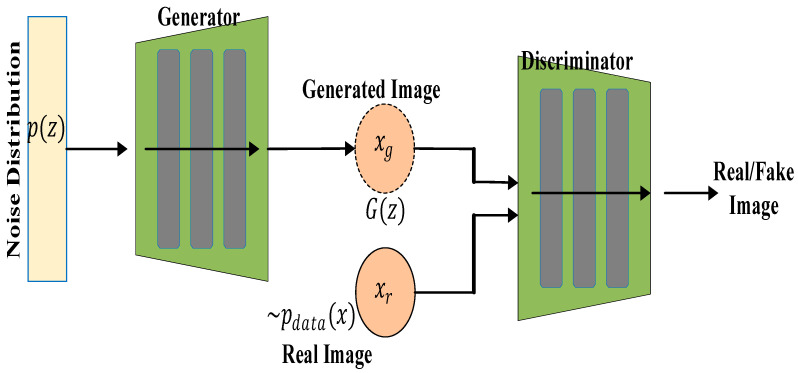
Structure of a GAN [79].

**Figure 8 diagnostics-15-01072-f008:**
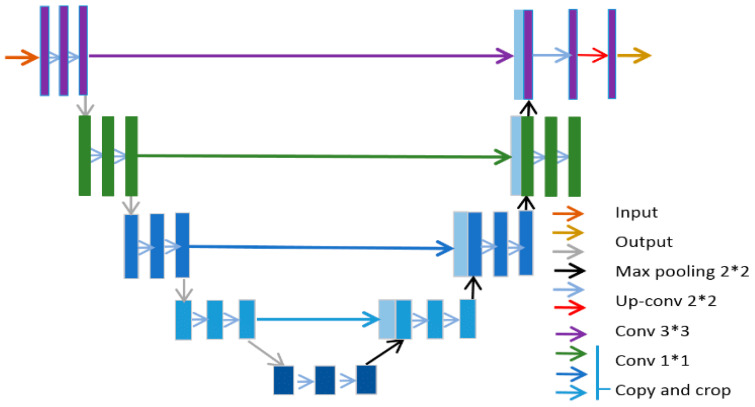
The architecture of U-Net [95].

**Figure 9 diagnostics-15-01072-f009:**
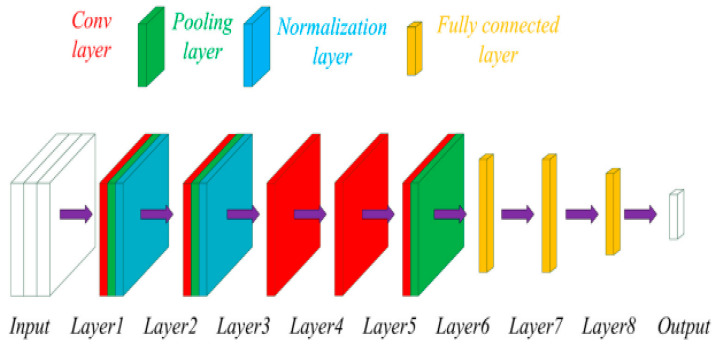
Structure of a transfer learning model—AlexNet [108].

**Figure 10 diagnostics-15-01072-f010:**
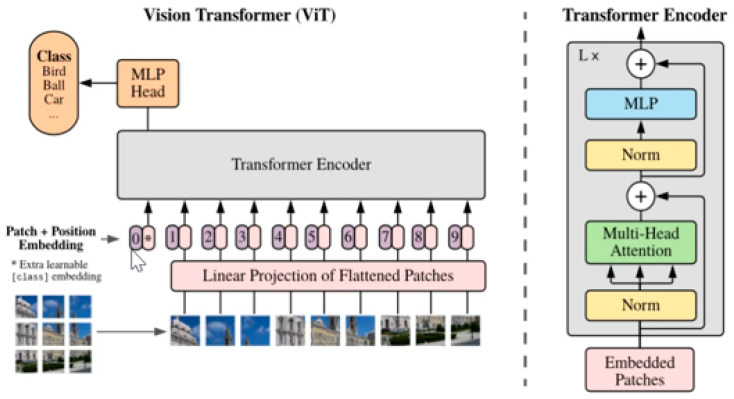
Vision transformer architecture [128].

**Table 1 diagnostics-15-01072-t001:** Recurrent neural network applications.

Article	Imaging Modality	Task	CNN Feature Extraction	Disease/Body Part	Variant Used
[58]	MRI	Classification	Y	Alzheimer’s	BGRU
[59]	Histopathological images	Classification	N	Breast cancer	None
[60]	MRI	Classification/segmentation	N	Brain tumour	LSTM
[57]	MRI	Segmentation	Y	Aorta	LSTM
[61]	MRI	Classification/localisation	Y	Knee ligament	LSTM
[62]	IRT	Classification	Y	Diabetes mellitus	LSTM
[56]	CT	Image denoising	N	Lungs	LSTM
[63]	MRI	Registration	Y	Brain cancer	LSTM

**Table 2 diagnostics-15-01072-t002:** Autoencoder-based techniques.

Article	Image Modality	Task	Disease/ Body Part
[67]	MRI	Augmentation/ segmentation	Brain
[69]	MRI	Denoising	Prostate
[70]	CT + others	Classification	Face
[71]	CT	Augmentation	Various
[72]	MRI and CT	Classification	Intracerebral haemorrhage
[73]	X-ray/digital histopathology	Anomaly detection	Various
[74]	Single-cell images	Classification	Myeloid leukaemia
[75]	None	Anomaly detection	None
[76]	CT	Classification	COVID-19
[77]	MRI	Denoising/classification	Autism/brain

**Table 3 diagnostics-15-01072-t003:** GAN-based techniques.

Article	Imaging Modality	Task	Disease/Body Part	Variant Used
[79]	MRI/retina fundus	Image synthesis	-	-
[82]	MRI	Image resolution	Brain	Cycle-GAN
[83]	CT	Image synthesis	COVID-19	Enhanced vanilla
[84]	X-ray/CT	Image Denoising	Chest/thorax	CGAN
[85]	Various	Image resolution	Various	Enhanced vanilla
[86]	-	Image synthesis	Skin cancer	DCGAN
[87]	MRI/CT	Image synthesis	Head/neck	Vanilla GAN
[88]	MRI/CT	Image resolution	Bladder cancer	Enhanced vanilla
[89]	Retina fundus/MRI	Image resolution	Various	Vanilla GAN
[90]	CT/MRI	Translation	Thorax/brain	CGAN

**Table 4 diagnostics-15-01072-t004:** U-Net segmentation techniques.

Article	Imaging Modality	Disease/Body Part	Variant Used
[95]	MRI	Brain tumour	Enhanced U-Net
[96]	Various	Various	Enhanced U-Net
[97]	CT	Hepatocellular carcinoma	Enhanced U-Net
[98]	CT	Liver	Enhanced U-Net
[99]	MRI	Brain tumour	None
[100]	Colour fundus	Diabetic retinopathy	Enhanced U-Net
[101]	MRI	Various/musculoskeletal	Enhanced U-Net
[102]	MRI	Lower limb muscle	Attention U-Net/SCU-Net
[103]	MRI	Musculoskeletal	Various
[104]	Various	Various	Enhanced U-Net (U-Net++)
[105]	Ultrasound	Breast cancer	Enhanced U-Net (attention gate)

**Table 5 diagnostics-15-01072-t005:** Summary of transfer learning techniques.

Article	Imaging Modality	Disease/Body Part	TL Variant/Best Model
[112]	Histopathological images	Breast cancer	ResNet 50
[113]	MRI	Brain tumour	Improved ResNet 50
[114]	MRI	Alzheimer’s	Various (EffiecientNet)
[115]	CT	Pulmonary nodules	Various (DenseNet)
[116]	X-ray/CT	COVID-19	Various (VGG 16)
[117]	MRI	Alzheimer’s	Modified ResNET 18
[118]	Ultrasound	Thyroid	VGG-16

## Data Availability

Not applicable.

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
