# Peer review of "Developments in Deep Learning Artificial Neural Network Techniques for Medical Image Analysis and Interpretation"

_diagnostics, 2025, doi:10.3390/diagnostics15091072_

Round 1
Reviewer 1 Report
Comments and Suggestions for Authors
This paper provides a comprehensive review of the literature regarding the use of deep learning neural network models for medical image analysis. The review is well structured. The main content of the review starts with a general and mathematical overview followed by a broad review of several pieces of research work.
The main criticisms of the paper include (a) ultrasound imaging as a main image modality has not been systematically reviewed like other medical image modalities although there are a few places where research on ultrasound were briefly mentioned; and (b) the Discussion section has not analysed the important issues relating to data availability, interpretability, overfitting and computational requirements. The current Discussion section serves as a summary of what has been described in the main body of the review. Although there can be good reasons for missing (a) due to the space constraint of the paper, such reasons must be clearly given. Otherwise, the paper gives an impression that a substantial amount of research work in ultrasound images for thyroid, breast, bladder, and ovarian cancers has simply been ignored. Regarding (b), this reviewer really hope that the revised version of the paper can spend the Discussion section on discussing the challenges as mentioned by the authors in the Abstract and review attempts in the literature for addressing those challenges.
Some minor points are summarized as follows:
- In the Abstract, please consider moving " - factors such as model trust ... remain ongoing concerns" into the conclusion and future work part because the article has not mentioned any research work in discussing any of those points.
- In the Introduction section, provide good reasons as why deep learning for ultrasound images is not included. Please also consider using a reference to existing reviews specifically on ultrasound imaging and deep learning. There are several in the literature already. This reviewer considers this change as a compulsory request.
- In Methods section, Figure 2 may cause some confusions over the excluded reports. Please ensure that the arrows start from the right places. Also in the Methods section, the angle of view from various deep learning architectures should be made clear. This review is not given from the clinical application angle. Since the journal may also be read by medical practitioners with strong interest in AI/ML, such a clarification is useful.
- In the Result section (Question for Editor: by the way, should the section have to be called with this title?),
- 3.1. Existing design of CNN for diagnostic purposes seems following three approaches (a) handcraft (or design from scratch), (b) transfer learning and (c) automatic search. Since the paper has a subsection on Transfer Learning, it should be made clear that this section reviews only the handcraft CNN architectures, and give reference to 3.6 for the transfer learning. It should also mention that there are research work in automatic search for CNN architectures.
- On page 10, top paragraph. the sentence "The ability of the neural network...." requires rephrasing.
- On page 18, Figure 9. It seems that a red layer shape for Conv Layer is missing from the top of the diagram (at least this is the case for the PDF version downloaded).
- On page 20, just before formula (35), "patches of size PP, such that" should be "patches of size P, such that".
- In the Discussion section, this reviewer does not agree with the use of "In this article, an in-depth discussion ..." in the first sentence. In fact, this review is a broad review and the discussion is also quite broad rather than in-depth so far.
- In the Discussion section, it is suggested that this section is rewritten to focus on research work in addressing some serious challenges faced by deep learning neural networks as the authors correctly mentioned in the Abstract. There is little need to repeat the descriptions already given in the main body of the review. This reviewer also considers this change as a "must".
Overall, the paper shows a good quality work.
Reviewer 2 Report
Comments and Suggestions for Authors
This review explored the transformative impact of deep learning algorithms on medical image analysis. Well-known methods, such as CNN, LSTM, GRU, U-Net, GAN, ViTs, etc., have been introduced for the medical image classification, segmentation, feature extraction, and diagnostics.
The formats of medical images are also investigated including CT, MRI, X-ray, etc. So this paper is valuable for the future study of the medical image analysis by the artificial intelligence technology.
To improve the contents of this paper, I suggest authors to add more state-of-the-art AI methods, such as LLMs and LVLMs, as follows.
1.arXiv:2402.14162v1 [cs.CV] 21 Feb 2024
2.10.26599/BDMA.2024.9020090
3.https://www.mercity.ai/blog-post/building-medical-ai-assistants-with-visual-llms
4. DOI: 10.21037/qims-23-892
Reviewer 3 Report
Comments and Suggestions for Authors
The article is a detailed literature review on medical image processing and deep learning. Before presenting the details of the existing studies in the literature, the studies were briefly reviewed and some studies were kept within the scope of the study. I have a few suggestions about the study.
1. Deep learning architectures such as CNN and Vision Transformer are summarized and the studies conducted in the medical field are summarized by giving tables. Which diseases or which types of images (CT, X-ray, etc.) should we use these architectures to classify? Add a table with a general summary to the discussion section.
2. The studies from 2025 were examined less in the article. Publications from 2025 should be included in the article. The latest technologies in medical image segmentation such as MedSAM and TransUNet should be evaluated. In addition to the ViT model, which has been superior to CNNs in classification problems, models such as Swin Transformer, and DeiT should be evaluated.
3. There are more successful architectures than Unet regarding segmentation. Why were only studies on Unet examined?
4. Although it seems appropriate to explain the layers and mathematical equations of the architectures, I do not know if it is necessary. These explanations are available in many articles or on internet platforms. Explanations for improved architectures would be more appropriate.
